# Effect of an Integrated Payment System on the Direct Economic Burden and Readmission of Rural Cerebral Infarction Inpatients: Evidence from Anhui, China

**DOI:** 10.3390/ijerph16091554

**Published:** 2019-05-03

**Authors:** Haomiao Li, Yingchun Chen, Hongxia Gao, Jingjing Chang, Dai Su, Shihan Lei, Di Jiang, Xiaomei Hu, Min Tan, Zhifang Chen

**Affiliations:** 1School of Medicine and Health Management, Tongji Medical College, Huazhong University of Science and Technology, Wuhan 430030, China; lihaomiao@hust.edu.cn (H.L.); gaohongxia@hust.edu.cn (H.G.); changjingjing@hust.edu.cn (J.C.); sudai@hust.edu.cn (D.S.); leishihan@hust.edu.cn (S.L.); jiangdi@hust.edu.cn (D.J.); huxiaomei@hust.edu.cn (X.H.); tanmin@hust.edu.cn (M.T.); chenzhifang@hust.edu.cn (Z.C.); 2Research Center for Rural Health Services, Hubei Province Key Research Institute of Humanities and Social Sciences, Wuhan 430030, China

**Keywords:** integrated payment system, cerebral infarction, direct economic burden, readmission, rural China

## Abstract

Rural China is piloting an integrated payment system, which prepays a budget to a medical alliance rather than a single hospital. This study aims to evaluate the effect of this reform on the direct economic burden and readmission rates of cerebral infarction inpatients. The settlement records of 78,494 cerebral infarction inpatients were obtained from the New Rural Cooperative Medical Scheme (NRCMS) database in Dingyuan and Funan Counties in the Anhui Province. The direct economic burden was estimated by total costs, out-of-pocket expenditures, the out-of-pocket ratio, and the compensation ratio of the NRCMS. Generalized additive models and multivariable linear/logistic regression were applied to measure the changes of the dependent variables along with the year. Within the county, the total costs positively correlated to the year (*β* = 313.10 in 2015; 163.06 in 2016). The out-of-pocket expenditures, out-of-pocket ratios, and the length-of-stay positively correlated to the year in 2015 (*β* = 105.10, 0.01, and 0.18 respectively), and negatively correlated to the year in 2016 (*β* = −58.40, −0.03, and −0.30, respectively). The odds ratios of the readmission rates were less than one within the county (0.70 in 2015; 0.53 in 2016). The integrated payment system in the Anhui Province has considerably reduced the direct economic burden for the rural cerebral infarction inpatients, and the readmission rate has decreased within the county. Inpatients’ health outcomes should be given further attention, and the long-term effect of this reform model awaits further evaluation.

## 1. Introduction

Provider payment system reform plays an important role in medical system reform. This reform, in which health insurance funds are transferred from the purchaser to healthcare providers [1], can considerably influence healthcare providers’ medical behavior, and can further influence the quality and efficiency of healthcare services as well as the enrollees’ direct economic burden and benefit level from health insurance [2,3]. Prospective payment system (PPS), which is gradually replacing post-payment systems around the world, has been proven effective for controlling spending by encouraging improvements to medical institutions’ efficiency [4,5]. A global budget (GB), which indicates that healthcare providers are provided an amount of money to spend with total flexibility on how and what to spend it on, is a representative type of PPS. GB is often incorporated by other payment methods (e.g., capitation and case payment) adopted by numerous countries [6]. GB can motivate medical institutions to contain the growth of medical expenditure with its inherent financial risk to assume excessive expenses [7]. However, GB performance regarding quality of care is less certain [4] and it may result in an inappropriate reduction of the examination or in prevarication for patients with critical illnesses or high treatment costs [8,9]. The reason for this is that the budget is often prepaid to a single hospital, subjecting the hospital to both high-medical and high-economic risks [10]. Thus, the relationship between medical insurance and hospitals is about regulation and being regulated and furthermore, the effect of the incentive mechanism of GB is not evident. To overcome this drawback, rural China has been piloting a relatively new payment method, in which the budget is prepaid to a medical community by capitation, rather than to a single medical institution. The medical community is an alliance of multiple interest-related hospitals. Moreover, capitation (for outpatients), fee-for-service (FFS, for inpatients), and case payment are incorporated into the GB system. This payment system was defined as an “integrated payment system”.

The Anhui Province is the first pilot area to implement this integrated payment system (since 2015) in China and has aimed at the rural healthcare delivery system. The ‘integration’ part of this provider payment system reform refers to not only the integration of different payment methods, but also to the integration of payment and health delivery systems. First, medical partnership (MP) within the county was constructed. The MP consists of a leading hospital (generally county-level hospital) and other members (township-level hospitals and village clinics). The MP is responsible for improving the health status of residents under its jurisdiction. Second, a GB was implemented into an MP. The medical insurance (the new rural cooperative medical system, NRCMS) sets 95% of the raised funds as the budget, after extracting risk-related funds, and transforms this budget to enrollees’ capitation. The total amount (lumpsum) is prepaid to the MP for the outpatient and inpatient services, necessary referrals, and reimbursement for the patients. Jurisdictional enrollees’ hospitalization services outside the county should be reimbursed by the MP. To motivate hospitals within the MP to provide services effectively, the balance of the budget can be reasonably distributed to the member hospitals in accordance with the suggested proportion of 6:3:1 (county-level hospital, township-level hospitals, and village clinics, respectively). Nevertheless, if the actual spending exceeds the budget, the MP should absorb the excess expenditure without being compensated by medical insurance. Third, different payment systems were implemented in different level hospitals and for different diseases. For example, case payment for hospitalization was implemented in county hospitals, and the selected clearly-diagnosed common diseases will be compensated by quota. Other complicated diseases are paid via FFS. GB for outpatient services was implemented in township hospitals and village clinics. The greatest advantage of this integrated payment system is that it translates the revenue of the hospitals within the MP to their costs, and the member hospitals’ awareness of cost-saving can be remarkably improved.

The integrated payment system in the Anhui Province has received considerable attention from scholars and health managers in China. Liu Shuang et al. evaluated the effect of payment reform in Dingyuan County, Anhui Province. They found that hospitalization expenses for patients who seek treatments outside the county are considerably higher than those incurred within the county [11]. Yu Yamin et al. selected Tianchang City (county-level city), Anhui Province, as the sample. They found that the expenses of inpatients and outpatients decreased in the People’s Hospital and the Traditional Chinese Medical Hospital, whereas the compensation ratio (CR) of medical insurance increased in both hospitals [12]. Nevertheless, existing studies about the payment system reform in Anhui have focused on its effect on patients’ distribution and medical expenditure, rather than the service quality. On the one hand, to control medical costs the member hospitals may reduce avoidable services and increase the service quality to improve patients’ health status. On the other hand, we cannot rule out the possibility that the member hospitals may decrease necessary services and service quality to reduce medical expenses.

In our study, we aimed to evaluate the integrated payment system reform in the Anhui Province comprehensively by concentrating on inpatients’ direct medical costs, the benefit level from medical insurance, and the quality of medical services. Service quality was mainly evaluated by readmission rates, which have great appeal as an indicator of hospitalization quality and service efficiency and are highly associated with medical costs [13,14,15,16,17,18]. Length of stay (LOS) was selected as another indicator to further prove if the change in readmission rates was appropriate. In addition, cerebral infarction inpatients were selected as the study population because as one of the most common clinical manifestations of a stroke, cerebral infarction not only has high morbidity rates, but also has heavy economic and health burdens on patients, society, and health delivery systems around the world [19,20]. In China, strokes are the third most common cause of death in rural areas, and the burden of strokes in China is more than twice the global average [21,22]. Besides, the readmission rates of cerebral infarction inpatients are higher than that of inpatients suffering from many other diseases, as the readmission is associated with the level of diagnosis and treatment, nursing, rehabilitation, and management within the hospitals, as well as medical insurance [23,24]. Thus, we believe that the readmission rates of cerebral infarction inpatients can reflect the service quality of the hospitals to a large extent, and the research on the disease burden of cerebral infarction is of great significance in the evaluation of the provider payment system. We hypothesized that the integrated payment system in the Anhui Province could reduce cerebral infarctions’ direct economic burden and improve service quality. Through the evaluation, we intended to demonstrate the applicability of this relatively new payment system in China and provide suggestions on optimization measures.

## 2. Materials and Methods

### 2.1. Data Source

Dingyuan and Funan Counties were selected as samples through typical sampling, because they were among the first batches of pilot locations in the Anhui Province in 2015. Dingyuan is located in the east of Anhui, whereas Funan is located in the northwest.

Data were obtained from the billing records of the NRCMS in Dingyuan and Funan Counties from 2014 to 2016. After extracting cases with missing data, a total of 78,494 (23,766 in 2014; 25,641 in 2015; 29,087 in 2016) hospitalization records of cerebral infarction inpatients were extracted for analysis. Cerebral infarction had the highest morbidity among all NRCMS inpatients (285,728 and 553,634 inpatients in Dingyuan and Funan, respectively) in the sample areas, which accounted for 7.48% and 10.34% in Dingyuan and Funan respectively, from 2014 to 2016. The payment method for cerebral infarction inpatients is FFS.

### 2.2. Study Variables

Cerebral infarction inpatients’ direct economic burden was estimated by their medical expenditures (total costs), actual costs (out-of-pocket [OOP] expenditures), and benefits from the NRCMS (out-of-pocket ratios [OOPRs] and CRs). The OOP referred to the actual cost paid by patients after medical insurance compensation. Service quality was estimated by the LOS and 30-day readmission rates (R30). These variables were defined as dependent variables, whereas “year” was defined as an independent variable to represent the time effects of the reform (2014 = before reform, 2015 = the reform year, 2016 = after the reform). Moreover, sex and age were introduced as covariates [25,26,27]. The indicators associated with disease severity, which may influence the dependent variables, could not be obtained from the database. However, the whole sample of cerebral inpatients was selected, which could overcome this limitation to a certain extent. Considering the change in price level, the total costs and OOP expenditures were adjusted in accordance with the consumer price index (CPI) in 2016.

### 2.3. Statistical Analysis

Smoothing splines generated in generalized additive models (GAMs) were used to explore the reform effect [28,29]. The model was as follows:(1)$$g\left(\mu \right)=\;s_{0}+s_{1}\left(X_{1}\right)+s_{2}\left(X_{2}\right)+\cdots +s_{p}\left(X_{p}\right)$$
(2)$$n=s_{0}+\sum_{i=1}^{p}s_{i}\left(X_{i}\right)$$

In Equation (1), $\mu =E(Y|X_{1},X_{2},\ldots ,X_{p})$; in Equation (2), *n* refers to linear predictive value, $s_{i}\left(.\right)$ refers to nonparametric smooth function, which can be smooth spline function, kernel function, or local regression smooth function, and its nonparametric form makes the model very flexible and can reveal the nonlinear effect of variables. The model does not require any assumptions of *Y* to *X*, and consists of the random component, the additive component, and the link function that joins the two. The distribution of the reaction variables is an exponential family, which can be binomial, Poisson, Gamma, and so on. We constructed a smooth curve on the basis of GAMs to fit the change of dependent variables intuitively under the change of year. Meanwhile, smooth curves fittings for the change of outcome variables within the county and out of county were conducted to reveal the different reform effects both within and out of the county. During the fitting process, the inpatients’ age and sex were controlled for.

Multivariable regressions were conducted to estimate the change of the dependent variables along with the year. The continuous outcome variables (total costs, OOP, CR, OOPR, and LOS) were performed as multivariable linear regressions, while the binary outcome variable (R30) was performed as a multivariable logistic regression [30,31]. As we focused on the reform effect, we set only one independent variable (year), with covariates (inpatients’ sex and age) controlled for. The linear trend was tested to confirm if there was a linear trend between the change of the independent variable *X* (generally the change of dummy variable after conversion) and the change of the dependent variable *Y*. Stratification regression analysis was conducted to compare the differences of the reform effects between inpatients within and outside the county. In stratified analysis, interactions were tested by log-likelihood ratio to estimate if the reform effect was significantly influenced by the variable “if patients were hospitalized within the county”.

## 3. Results

### 3.1. Sample Characteristics

Table 1 shows the basic characteristics of cerebral infarction inpatients under the NRCMS of Dingyuan and Funan Counties from 2014 to 2016. The number of male inpatients was higher than that of female inpatients in 2014 and 2015, but this was reversed in 2016. The majority of the inpatients were slightly older (60 to 74 years old), which constituted over 40% of the total sample and over 50% in 2015. The constituent ratio of cerebral infarction inpatients hospitalized within the county increased annually, whereas the constituent ratio of those outside the county decreased. The means of LOS, the total cost, and OOP expenditure increased in 2015, but decreased in 2016. The OOPR decreased in 2016, whereas the CR increased in 2016. The readmission rates decreased annually. All of the differences were statistically significant (*p* < 0.001).

### 3.2. The Direct Economic Burden and Service Quality of the Whole Sample

Figure 1 shows the smooth curves fitting for the changes of dependent variables along with year (with sex and age of inpatients controlled for). The total costs, OOP expenditures, OOPR, and LOS increased in 2015, but decreased in 2016. The change of CR was opposite to that of OOPR. The R30 decreased year by year. Similar results were obtained through multivariable regression models (Table 2). After the sex and age were adjusted, the total costs positively correlated to the year (*β* = 258.21 in 2015 and 83.97 in 2016). A positive correlation existed between the year and OOP expenditures (*β* = 103.45), OOPR (*β* = 0.01), and LOS (*β* = 0.10) in 2015, and the correlation changed to negative in 2016 (*β*: OOP expenditures = −79.49, OOPR = −0.03, LOS = −0.38). The correlation between the year and CR was negative in 2015 (*β* = −0.01) and positive in 2016 (*β* = 0.03). The correlation between the year and R30 was negative (odds ratio (OR) = −0.31 in 2015 and −0.56 in 2016). The tests for linear trend were significant in OOP expenditures, OOPR, CR, LOS, and R30 (*p* < 0.001).

### 3.3. The Direct Economic Burden and Service Quality of Cerebral Infarction Inpatients Within and Outside the County

To estimate the reform effect further, cerebral infarction inpatients were divided into two groups in accordance with the institutions they were hospitalized in: within and outside the county. Figure 2 shows the smooth curve fitting for the changes of dependent variables along with the year among inpatients both within and outside the county, with sex and age controlled for. The total costs, OOP, and LOS increased slightly in 2015 and decreased in 2016 within the county, whereas they kept increasing out of the county. The CR increased in 2016 within the county, whereas it decreased in 2015 and remained stable in 2016 out of the county. The change of the OOPR was opposite to that of the CR. The R30 kept decreasing within the county, whereas it was the total opposite out of the county. The hierarchical multivariable regression models (Table 3) show that the total costs positively correlated to the year within the county (*β* = 313.10 in 2015, *β* = 163.06 in 2016) and outside the county (*β* = 3299.79 in 2015, *β* = 3663.67 in 2016). The OOP expenditures, OOPR, and LOS positively correlated to the year in 2015 within the county (*β*: OOP expenditures = 105.10, OOPR = 0.01, LOS = 0.18) and outside the county (*β*: OOP expenditures = 2793.37, OOPR = 0.19, LOS = 0.39). However, they negatively correlated to the year in 2016 within the county (*β*: OOP expenditures = −58.40, OOPR = −0.03, LOS = −0.30) and still positively correlated to the year in 2016 out of the county (*β*: OOP expenditures = 3001.65, OOPR = 0.18, LOS = 0.42). The CR negatively correlated to the year in 2015 (β = −0.01) within the county and positively correlated to the year in 2016 (*β* = 0.03) within the county, whereas it kept negatively correlating to the year out of the county (*β* = −0.19 in 2015 and −0.18 in 2016). The ORs of the R30 were less than one within the county (OR = 0.70 in 2015 and 0.53 in 2016) and more than one outside the county (OR = 1.33 in 2015 and 2.00 in 2016). The interactions between all of the dependent variables and the year were substantial (*p* < 0.001), which proved that the relationships were tortured by the variable “if inpatients were hospitalized within the county” and further proved that the reform takes effect within the county.

## 4. Discussion

GAMs were utilized in our study and the smooth curves fittings for GAMs were introduced to visually represent the reform effects. Smooth curves fittings for GAMs of different groups were constructed to further prove the reform effect. Tests for linear trends and interactions were used to supplement the estimation, as the former can scientifically estimate if dependent variables’ changing trends along with the year are statistically significant, and the latter can identify the reform effect between different groups [32,33]. The results of multivariable regressions were similar to the results of GAMs, which not only corroborated the applicability of GAMs but also revealed the directions of the change for the core indexes under the typical reform.

Under the integrated payment system, the direct economic burden of cerebral infarction inpatients has been reduced, which is mainly reflected in the decreasing trend of OOP expenditures after the reform within the county. Moreover, the benefits from medical insurance for inpatients within the county has been improved, as the CR increased and the OOPR decreased. In sample areas, GB is an effective payment method for improving medical institutions’ consciousness of cost-saving, as it can convert medical insurance funds from medical revenue to medical cost [6,34]. More importantly, the budget is prepaid to the MP, which changes the relationship of interests among the member hospitals. This mode is similar to the “two-sided” mode of the Medicare Shared Savings Program established under the Affordable Care Act in the US. Under this mode, the provider groups can not only share savings with Medicare if the group’s spending is below its pre-specified target, but also assume the risk for excess spending over their targets [35,36]. Similarly, the integrated payment system in the Anhui Province can impel the member hospitals to save the NRCMS funds from several different aspects. First, the member institutions should reduce the provisions of unnecessary medical services, such as inappropriate admissions, medical examinations, and drug uses. This condition can be reflected in the shortened LOS among inpatients within the county, which has been proven to be a predominant response to PPS [37,38,39]. Second, the member institutions will proactively ameliorate service quality to improve the jurisdictional enrollees’ health status by strengthening medical capacity. After saving medical insurance funds, the reimbursement level of medical insurance for inpatients can be considerably improved.

Furthermore, the readmission rates of cerebral infarction inpatients decreased within the counties. Readmission rates have been proven to be a core index for screening the quality of care by payers (e.g., Medicare) and multi-hospital systems (e.g., the Department of Veterans Affairs) [13], because high readmission rates can be a sign of poor clinical care and can indicate a patient’s worsening condition [40], further causing a high economic burden for the patients [41]. Heggestad found that admissions to hospitals with relatively short average LOS have a considerably high risk of 30-day readmission [42]. In this study, we found that under a shortened LOS, the readmission rates decreased. This may be due to the supporting assessment mechanism associated with controlling LOS and readmission rates, as this assessment will influence the distribution of the balance. In addition, the improvement of quality will effectively affect the health status of residents and further increase the amount of the balance.

Although the reform is effective in the sample areas, we still cannot ignore the underlying problems. The improvement of health outcomes for the inpatients is the ultimate goal of the reform in the Anhui Province, as well as for all healthcare reforms. Hospitals within the counties should pay further attention to the continuity of medical services. Thus, they should provide prevention, diagnosis, treatment, referral, and rehabilitation services for inpatients so that patients can receive the most appropriate services in the most appropriate institutions at minimal costs.

## 5. Limitations

This study has several limitations. First, the sample only includes cerebral infarction inpatients, which cannot represent all inpatients that may be influenced by the reform at different levels. Second, our data cover 2014 to 2016, however, the policy effects may be delayed and may not have been totally manifested in our study phase. Third, some confounding factors that influence the reform effect may not be included in the database, such as disease severity and service capability of medical institutions. Fourth, the appropriateness of service processes in hospitals could not be effectively evaluated due to the limitation of the database. Finally, we could not assess the reform effect on residents’ actual health outcomes in our study, not only because the variables in the NRCMS database are limited but also because a thorough scientific index system that estimates the health outcomes with MP in China is unavailable.

## 6. Conclusions

Limited studies analyze the effect of the integrated payment system, as a relatively new payment system reform mode in China, especially from the aspect of medical service quality. Our study not only demonstrated that the integrated payment system is effective in reducing the direct economic burden to inpatients and improving the benefit level of medical insurance, but also proves that the reform can optimize the quality of medical services. We believe that the integrated payment system is feasible and promising in rural China. We also suggest that further attempts to focus on the health outcomes of residents should be made in the next reform process, as improving health status is the ultimate goal of all health reforms.

## Figures and Tables

**Figure 1 ijerph-16-01554-f001:**
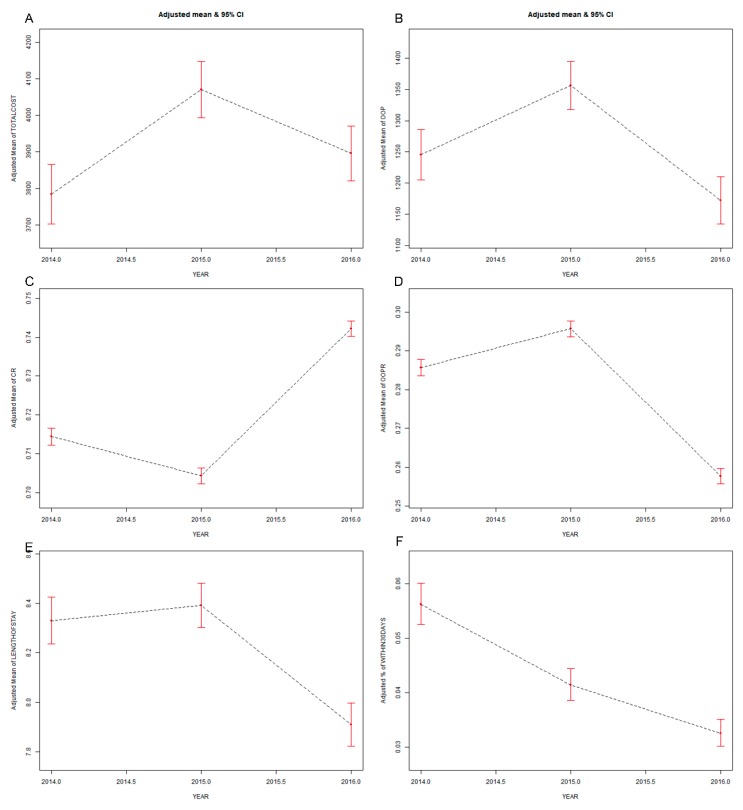
Direct economic burden and service quality for the whole sample. Ordinate: (**A**) adjusted mean of total costs; (**B**) adjusted mean of out-of-pocket (OOP) expenditures; (**C**) adjusted mean of the compensation ratio (CR); (**D**) adjusted mean of the OOP ratio; (**E**) adjusted mean of length of stay; (**F**) adjusted mean of the 30-day readmission (R30) rate.

**Figure 2 ijerph-16-01554-f002:**
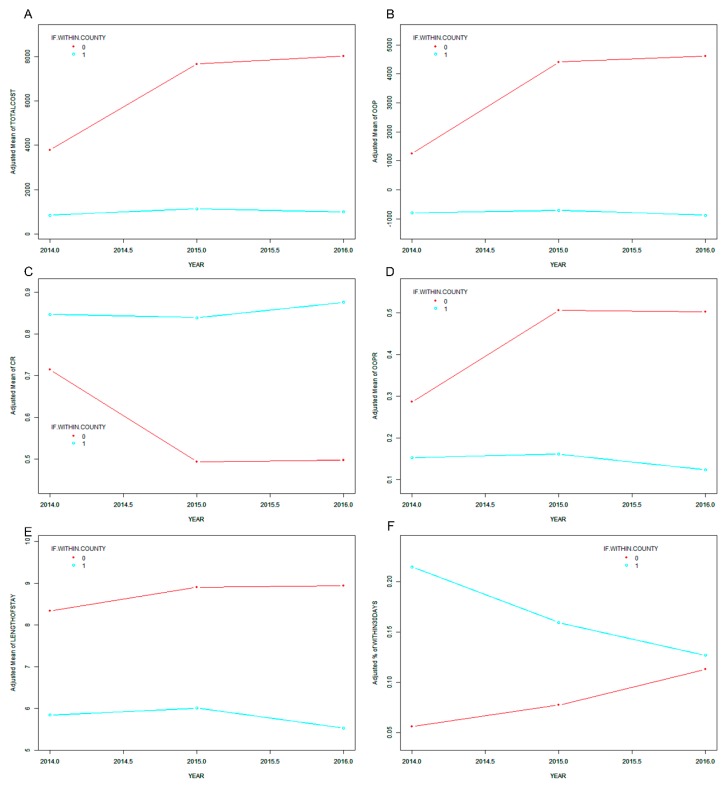
Direct economic burden and service quality for cerebral infarction inpatients within and outside the county (0: out of the county, 1: within the county). Ordinate: (**A**) adjusted mean of total costs; (**B**) adjusted mean of OOP expenditures; (**C**) adjusted mean of CR; (**D**) adjusted mean of the OOP ratio; (**E**) adjusted mean of length of stay; (**F**) adjusted mean of the R30 rate.

**Table 1 ijerph-16-01554-t001:** Basic characteristics of cerebral infarction inpatients under the new rural cooperative medical system (NRCMS) in Dingyuan and Funan County, 2014–2016.

	2014	2015	2016	*p*-Value
***N***	23,766	25,641	29,087	
**Sex**				<0.001
Male	12,364 (52.02%)	13,146 (51.27%)	14,468 (49.74%)	
Female	11,402 (47.98%)	12,495 (48.73%)	14,619 (50.26%)	
**Age group**				<0.001
<45	2466 (10.38%)	630 (2.46%)	742 (2.55%)	
45–59	4096 (17.23%)	4669 (18.21%)	5714 (19.64%)	
60–74	10,584 (44.53%)	12,858 (50.15%)	14,516 (49.91%)	
≥75	6620 (27.85%)	7484 (29.19%)	8115 (27.90%)	
**Institution level**				<0.001
Within the county	21,294 (89.60)	24,646 (96.12%)	28,132 (96.72%)	
Outside the county	2472 (10.40%)	995 (3.88%)	955 (3.28%)	
**Length of stay (day)**	8.33 ± 6.72	8.46 ± 5.25	7.98 ± 5.26	<0.001
**Total costs (¥)**	3784.52 ± 5096.17	4023.56 ± 4959.98	3853.25 ± 4808.94	<0.001
**OOP expenditures (¥)**	1245.44 ± 2532.52	1325.20 ± 2525.80	1145.21 ± 2382.07	<0.001
**OOPR**	0.29 ± 0.13	0.29 ± 0.14	0.26 ± 0.12	<0.001
**CR**	0.71 ± 0.13	0.71 ± 0.14	0.74 ± 0.12	<0.001
**30-day readmission**				<0.001
No	22,430 (94.38%)	24,521 (95.63%)	28,099 (96.60%)	
Yes	1336 (5.62%)	1120 (4.37%)	988 (3.40%)	

Note: Data in the table: Mean ± standard deviation/number (constituent ratio, %). The test for continuous variables was independent samples *t*-test, and the test for categorical variables was Chi-squared test.

**Table 2 ijerph-16-01554-t002:** Results of multivariable linear/ logistic regression models for the whole sample.

	Non-Adjusted		Adjusted	
	*β* or OR (95%CI)	*p* Trend	*β* or OR (95%CI)	*p* Trend
**Total costs (¥) & year**				
2014	0	0.201	0	0.118
2015	239.04 (151.74, 326.34)		258.21 (170.38, 346.04)	
2016	68.73 (−16.04, 153.51)		83.97 (−1.25, 169.18)	
**OOP expenditures (¥) & year**				
2014	0	<0.001	0	<0.001
2015	79.76 (36.07, 123.45)		103.45 (59.51, 147.38)	
2016	−100.22 (−142.65, –57.80)		−79.49 (−122.12, –36.86)	
**CR & year**				
2014	0	<0.001	0	<0.001
2015	−0.01 (−0.01, −0.01)		−0.01 (−0.01, −0.01)	
2016	0.03 (0.03, 0.03)		0.03 (0.03, 0.03)	
**OOPR & year**				
2014	0	<0.001	0	<0.001
2015	0.01 (0.01, 0.01)		0.01 (0.01, 0.01)	
2016	−0.03 (−0.03, −0.03)		−0.03 (−0.03, −0.03)	
**LOS & year**				
2014	0	<0.001	0	<0.001
2015	0.13 (0.03, 0.23)		0.10 (−0.01, 0.20)	
2016	−0.35 (−0.45, −0.25)		−0.38 (−0.48, −0.28)	
**R30 & year**				
2014	1	<0.001	1	<0.001
2015	−0.27 (−0.35, −0.18)		−0.31 (−0.40, −0.23)	
2016	−0.53 (−0.61, −0.44)		−0.56 (−0.65, −0.48)	

Note: The table shows the regression coefficients of different outcome variables & the independent variable (year). The regression coefficient *β* was for continuous outcome variables (total costs, OOP, CR, OOPR, and LOS), while OR was for binary outcome variables (R30). Adjusted: sex and age of the inpatients were adjusted. The year “2014” was set as the dummy variable.

**Table 3 ijerph-16-01554-t003:** Hierarchical regression results of cerebral infarction inpatients within and outside the county.

	Within the County (*N* = 74,072)	Out of the County (*N* = 4422)	*p* for Interaction
**Total costs (¥) & year**			
2014	0	0	<0.001
2015	313.10 (238.48, 387.73)	3299.79 (2408.25, 4191.33)	
2016	163.06 (90.56, 235.56)	3663.67 (2763.31, 4564.02)	
**OOP expenditures (¥) & year**			
2014	0	0	<0.001
2015	105.10 (73.47, 136.73)	2793.37 (2278.95, 3307.79)	
2016	−58.40 (−89.13, −27.67)	3001.65 (2482.15, 3521.16)	
**CR & year**			
2014	0	0	<0.001
2015	−0.01 (−0.01, −0.01)	−0.19 (−0.20, −0.17)	
2016	0.03 (0.03, 0.03)	−0.18 (−0.20, −0.17)	
**OOPR & year**			
2014	0	0	<0.001
2015	0.01 (0.01, 0.01)	0.19 (0.17, 0.20)	
2016	−0.03 (−0.03, -0.03)	0.18 (0.17, 0.20)	
**LOS & year**			
2014	0	0	<0.001
2015	0.18 (0.08, 0.28)	0.39 (−0.34, 1.11)	
2016	−0.30 (−0.40, −0.20)	0.42 (−0.31, 1.15)	
**R30 & year**			
2014	1	1	<0.001
2015	0.70 (0.64, 0.76)	1.33 (0.73, 2.43)	
2016	0.53 (0.49, 0.58)	2.00 (1.16, 3.45)	

Note: The table shows the regression coefficients of different outcome variables and the independent variable (year). The regression coefficient *β* was for continuous outcome variables (total costs, OOP, CR, OOPR, and LOS), while OR was for binary outcome variables (R30). The year “2014” was set as the dummy variable.

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
