# Peer review of "Effect of an Integrated Payment System on the Direct Economic Burden and Readmission of Rural Cerebral Infarction Inpatients: Evidence from Anhui, China"

_ijerph, 2019, doi:10.3390/ijerph16091554_

Round 1
Reviewer 1 Report
This paper uses generalized additive models and multiple linear regression to evaluate the effect of integrated payment on the direct economic burden and readmission of rural cerebral infarction inpatients. The content is significance for shaping and evaluating public policy. But the methods describing needs more detail, especially for multiple linear regression models. The figures and tables in results need more explanation.
Author Response
Dear Reviewer,
Thanks for your kind commending for this paper and your valuable suggestions. Our responses to your comments are below:
Comment 1: The methods describing needs more detail, especially for multiple linear regression models.
Response: It is very good suggestions and we agree it is necessary. We have made more description on the methods.
Page 4 Line 152-166: Meanwhile, smooth curves fitting for the change of outcome variables within the county and out of county were conducted, to reveal the different reform effects within the county and out of the county. During the fitting process, the inpatients’ age and sex were controlled.
Multivariable regressions were conducted to estimate the change of the dependent variables along with year. The continuous outcome variables (Total costs, OOP, CR, OOPR and LOS) were performed as multivariable linear regressions, while binary outcome variable (R30) was performed as multivariable logistic regression [30-31]. As we focused on the reform effect, we just set only one independent variable (year), with covariates (inpatients’ sex and age) controlled. The linear trend was tested to confirm if there was a linear trend between the change of independent variable X (generally the change of dummy variable after conversion) and the change of dependent variable Y. Stratification regression analysis was conducted to compare the differences of the reform effects between inpatients within and outside the county. In stratified analysis, interactions were tested by log-likelihood ratio, to estimate if the reform effect was significantly influenced by the variable “if patients hospitalized within the county”.
Comment 2: The figures and tables in results need more explanation.
Response: Thanks for your valuable suggestions. We have made more explanations in the results section.
Page 4 Line 176-177: All the differences were statistically significant (P < 0.001).
Page 5 Line 184-187: Figure 1 shows the smooth curves fitting for the changes of dependent variables along with year (with sex and age of inpatients controlled). The total costs, OOP expenditures, OOPR and LOS increased in 2015, but decreased in 2016. The change of CR was opposite to OOPR. The R30 decreased year by year.
Page 7 Line 211-216: The total costs, OOP and LOS increased slightly in 2015 and decreased in 2016 within the county, whereas they kept increasing out of the county. The CR increased in 2016 within county, whereas it decreased in 2015 and kept stable in 2016 out of the county. The change of the OOPR was opposite to the CR. The R30 kept decreasing within the county, whereas it was totally opposite out of the county.
Page 7 Line 220-226: However, they negatively correlated to the year in 2016 within the county (β: OOP expenditures = −58.40; OOPR = −0.03; LOS = −0.30) and still positively correlated to the year in 2016 out of the county (β: OOP expenditures = 3001.65; OOPR = 0.18; LOS = 0.42). The CR negatively correlated to the year in 2015 (β = −0.01) within the county and positively correlated to the year in 2016 (β = 0.03) within the county, whereas it kept negatively correlating to the year out of the county (β = −0.19 in 2015 and −0.18 in 2016).
Reviewer 2 Report
The present study was to examine the effect of this reform on cerebral infarction inpatients’ direct economic burden and readmission. Authors suggested that the integrated payment system in Anhui Province has considerably reduced the rural cerebral infarction inpatients’ direct economic burden, with the readmission rate decreased within the county. Inpatients’ health outcomes should be paid further attention, and the long-term effect of this reform mode awaits further evaluation.
This manusctipt will be published after revision.
1) What is the hypothesis of this study? It need to be specific.
2) Figures are hard to see for readers, so it need to revise.
3) For English, proofreading will be necessary.
Author Response
Dear Reviewer,
Thanks for your kind commending for this paper and your valuable suggestions. We are pleased to response to them point by point and changes in the article. Our responses to your comments are as follows:
Comment 1: What is the hypothesis of this study? It need to be specific.
Response: It is a very good question. In this study, we hypothesized that the integrated payment system in Anhui Province could reduce cerebral infarctions’ direct economic burden and improve service quality. We added this hypothesis in Page 3 Line 110-112.
Comment 2: Figures are hard to see for readers, so it need to revise.
Response: Thanks for your valuable suggestions. We have added more explanations about the variables and statistical methods below each table (as “Note:”, and we added more explanations to describe these tables more clearly in the results section.
Table 1: Note: ①Data in the table: Mean ± standard deviation/ Number (constituent ratio, %).
②The test for continuous variables is independent samples t-test, and the test for categorical variables is Chi-squared test.
Table 2: Note: ①The table shows the regression coefficients of different outcome variables & the independent variable (year). The regression coefficient β was for continuous outcome variables (Total costs, OOP, CR, OOPR and LOS), while OR was for binary outcome variables (R30).
②Adjusted: sex and age of the inpatients were adjusted.
③The year “2014” was set as the dummy variable.
Table 3: Note: The table shows the regression coefficients of different outcome variables & the independent variable (year). The regression coefficient β was for continuous outcome variables (Total costs, OOP, CR, OOPR and LOS), while OR was for binary outcome variables (R30). The year “2014” was set as dummy variable.
Comment 3: For English, proofreading will be necessary.
Response: Thanks for your valuable suggestions. With the help of a native English speaking professional, we have revised the manuscript word by word. The details were shown in the revised manuscript.
Round 2
Reviewer 2 Report
This manuscript is revised based on the reviewers comments, so I recommend to publish this.